# Investigation of water adsorption and hygroscopicity of atmospherically relevant particles using a commercial vapor sorption analyzer

Wenjun Gu[1,4], Yongjie Li[2], Jianxi Zhu[3], Xiaohong Jia[1,4], Qinhao Lin[1], Guohua Zhang[1], Xiang Ding[1], Wei Song[1], Xinhui Bi[1], Xinming Wang[1,5], Mingjin Tang[1,*]

State Key Laboratory of Organic Geochemistry and Guangdong Key Laboratory of Environmental Protection and Resources Utilization, Guangzhou Institute of Geochemistry, Chinese Academy of Sciences, Guangzhou 510640, China
Department of Civil and Environmental Engineering, Faculty of Science and Technology, University of Macau, Avenida da Universidade, Taipa, Macau, China
CAS Key Laboratory of Mineralogy and Metallogeny and Guangdong Provincial Key Laboratory of Mineral Physics and Material Research & Development, Guangzhou Institute of Geochemistry, Chinese Academy of Sciences, Guangzhou 510640, China
University of Chinese Academy of Sciences, Beijing 100049, China
Center for Excellence in Regional Atmospheric Environment, Institute of Urban Environment, Chinese Academy of Sciences, Xiamen 361021, China

Correspondence: M. J. Tang (Email: mingjintang@gig.ac.cn)

**Abstract**

Water adsorption and hygroscopicity are among the most important physicochemical properties of aerosol particles, largely determining their impacts on atmospheric chemistry, radiative forcing, and climate. Measurements of water adsorption and hygroscopicity of nonspherical particles under subsaturated conditions are non-trivial because many widely used techniques require the assumption of particle sphericity. In this work we describe a method to directly quantify water adsorption and mass hygroscopic growth of atmospheric particles for temperature in the range of 5-30 °C, using a commercial vapor sorption analyzer. A detailed description of instrumental configuration and experimental procedures, including relative humidity (RH) calibration, is provided first. It is then demonstrated that for $(NH_4)_2SO_4$ and NaCl, deliquescence relative humidities (DRHs) and mass hygroscopic growth factors measured using this method show good agreements with experimental and/or theoretical data from literature. To illustrate its ability to measure water uptake by particles with low hygroscopicity, we used this instrument to investigate water adsorption by $CaSO_4 \cdot 2H_2O$ as a function of RH at 25 °C. The mass hygroscopic growth factor of $CaSO_4 \cdot 2H_2O$ at 95% RH, relative to that under dry conditions (RH < 1%), was determined to be (0.450±0.004)% (1 σ). In addition, it is shown that this instrument can reliably measure a relative mass change of 0.025%. Overall, we have demonstrated that this commercial instrument provides a simple, sensitive and robust method to investigate water adsorption and hygroscopicity of atmospheric particles.

## 1 Introduction

Atmospheric aerosol particles, directly emitted by natural and anthropogenic processes or secondarily formed in the atmosphere, have significant impacts on air quality, visibility, human health, and radiative and energy balance of the Earth system (Pöschl, 2005; Seinfeld and Pandis, 2006). The ability to uptake water is among the most important physicochemical properties of aerosol particles, and it largely determines their impacts on atmospheric chemistry and climate (Martin, 2000; Rubasinghege and Grassian, 2013; Farmer et al., 2015; Tang et al., 2016). The ability of aerosol particles to uptake water depends on particle composition, relative humidity (RH), and temperature (Martin, 2000; Tang et al., 2016). Under subsaturated conditions (RH <100%), the ability to uptake water is typically called water adsorption in surface science and hygroscopicity in aerosol science (Tang et al., 2016). Under supersaturated conditions, aerosol particles can be activated to cloud droplets (McFiggans et al., 2006; Petters and Kreidenweis, 2007) and ice particles if temperature is below 0 ºC (Pruppacher and Klett, 1994; Vali et al., 2015).

Hygroscopicity of atmospheric particles has been extensively investigated by a large number of studies, and many experimental techniques have been developed. These techniques have been summarized and discussed by a very recent review paper (Tang et al., 2016), and here we only mention widely used ones. For airborne monodisperse particles typically produced by a differential mobility analyzer (DMA), the hygroscopicity can be determined by measuring their diameters at dry (typically at RH <15% or lower) and humidified conditions (Swietlicki et al., 2008; Freedman et al., 2009; Robinson et al., 2013; Lei et al., 2014). Typically, the diameter change is determined by using a scanning particle mobility sizer (in which mobility diameters are measured) (Vlasenko et al., 2005; Swietlicki et al., 2008; Herich et al., 2009; Koehler et al., 2009; Wu et al., 2011) or aerosol extinction-cavity ring down spectrometry (in which optical diameters are measured) (Freedman et al., 2009; Attwood and

Greenslade, 2011). These techniques require an underlying assumption that particles are
spherical. However, a few important types of particles in the troposphere, including mineral
dust and soot, are known to be non-spherical (Veghte and Freedman, 2012; Ardon-Dryer et al.,
2015). Therefore, although these techniques can provide useful information, it is difficult to
quantitatively determine the amounts of water associated with non-spherical particles at a given
RH (Tang et al., 2016). Single particle levitation techniques, which measure light scattering
intensity to determine the size (and thus the hygroscopic growth) of levitated particles, also
have similar drawbacks (Krieger et al., 2012).
There are several techniques which can be applied to quantify the amount of water
associated with non-spherical particles at given temperature and RH. For example, adsorbed
water can be measured by FTIR by its IR absorption at around 3400 and 1645 cm$^{-1}$ (Goodman
et al., 2001; Frinak et al., 2005; Ma et al., 2010a). However, it is non-trivial to convert IR
absorption intensity to the amount of adsorbed water (Schuttlefield et al., 2007a; Tang et al.,
2016). Several previous studies have used quartz crystal microbalance (QCM) to measure DRH
and mass hygroscopic growth of particles (Schuttlefield et al., 2007b; Arenas et al., 2012; Liu
et al., 2016; Yeşilbaş and Boily, 2016). The frequency change of the quartz crystal in a QCM,
according to the Sauerbrey equation, is proportional to the change in mass of particles loaded
on the crystal (Schuttlefield et al., 2007b). In addition, the amount of water associated with
particles at a given RH can also be determined by measuring the change in water vapor pressure
before and after exposure of particles to water vapor (Ma et al., 2010b; Ma et al., 2012), in a
manner similar to determination of Brunauer-Emmett-Teller surface area. In theory the
electrodynamic balance can be used to investigate the mass hygroscopic growth of non-
spherical particles (Chan et al., 2008; Lee et al., 2008; Pope, 2010; Griffiths et al., 2012). To
our knowledge, however, this technique has not been applied to mineral dust and soot particles
yet.
In this work we have developed an experimental method to investigate water adsorption
and hygroscopicity of atmospheric particles, using a vapor sorption analyzer which is
commercially available. We note that two groups have used similar techniques to measure
water adsorption by $CaCO_3$ and Arizona Test dust (Gustafsson et al., 2005) and DRH of
malonic acid, sodium oxalate and sodium malonate (Beyer et al., 2014; Schroeder and Beyer,
2016). Nevertheless, the performance of this technique has never been systematically evaluated.
To validate this experimental method, we have determined DRHs of six compounds as a
function of temperature from 5 to 30 $^{\circ}C$, and the measured DRHs, varying from ~20% to ~90%
RH, show excellent agreement with literature values. In addition, mass hygroscopic growth
factors (MGF) of $(NH_4)_2SO_4$ and NaCl have been measured as a function of RH at 25 and 5 $^{\circ}C$,
and the measured MGF agree very well with those predicted by the E-AIM model
(http://www.aim.env.uea.ac.uk/aim/aim.php; last accessed: 11 January 2017). Detailed
description of the E-AIM model can be found elsewhere (Clegg et al., 1998; Friese and Ebel,
2010). Hygroscopic growth factors, calculated using the E-AIM model, has been widely used
to compare with experimental measurements to verify the performance of a variety of
instruments, techniques and/or methods developed for hygroscopic growth studies (Pope et al.,
2010; Lei et al., 2014; Estillore et al., 2016). We show that this instrument can measure a
relative mass change (due to water uptake) of <0.025% within 6 hours and <0.05% within 24
hours, and the accuracy of mass change measurement is mainly limited by baseline drifts.
These features make this instrument particularly useful for laboratory studies of water
adsorption by nonspherical particles and/or particles with low hygroscopicity.

## 113 2 Experimental section

The instrument used in this work is a vapour sorption analyser (Q5000SA) manufactured
by TA Instruments (New Castle, DE, USA). The first part of this section provides a general
description of this instrument, and the second part describes experimental methods used in this
work.

## 2.1 Instrument description

Figure 1 shows the schematic diagram of the vapor sorption analyzer used in this work to
measure hygroscopicity and water adsorption of particles of atmospheric relevance. This
instrument consists of two main parts: 1) a high-precision balance used to measure the mass of
samples; 2) a humidity chamber in which temperature and RH can be precisely regulated and
also monitored online.

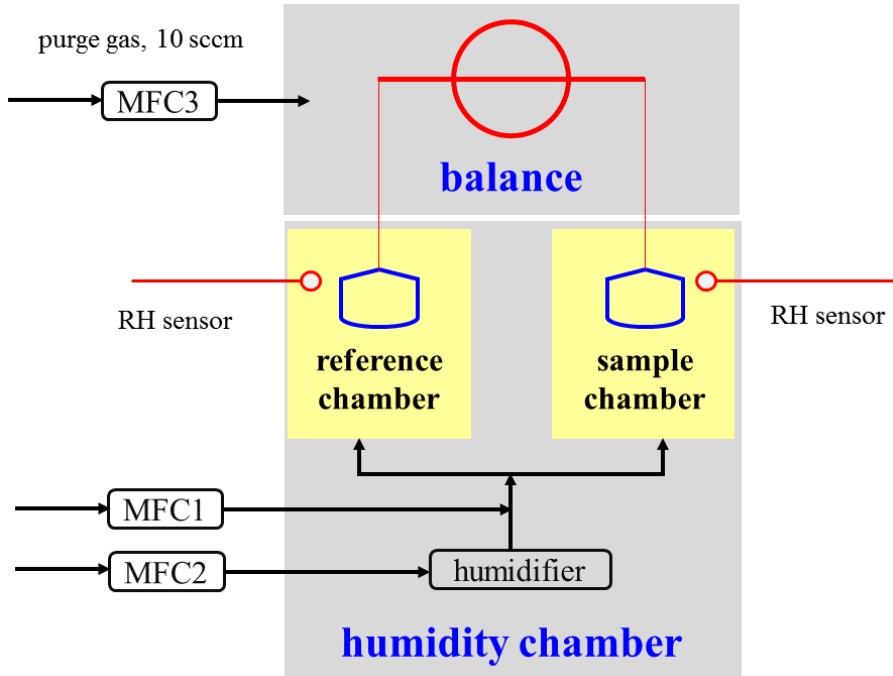


**Figure 1.** Schematic diagram of Q5000SA used in this work. MFC: mass flow controller. High

purity $N_2$ is used for all the three gas flows regulated by MFC1, MFC2, and MFC3, respectively.

### 2.1.1 High-precision balance

The balance simultaneously measures the mass of an empty pan (serving as a reference)
and a sample pan which contains particles under investigation. Each pan is connected to the
balance by a hang-down wire which has a hook at the lower end to hold the pan. The balance
is housed in a chamber which is temperature regulated. To avoid moisture condensation, the
balance chamber is purged with a 10 sccm (standard cubic centimetre per min) $N_2$ flow
regulated by a mass flow controller (MFC3).
The balance has a dynamic range of 0-100 mg. Typical dry mass of particles used in our
experiments are in the range of 1-10 mg so that the total mass of particles, due to adsorption of
water at high RH, does not exceed the upper limit of mass range. Powdered particles are
transferred into the sample pan using a small stainless-steel spatula. The mass of the sample
would not affect the measured mass ratio of dry particles to associated water under a given
condition; however, it would take more time to reach the equilibrium if the sample mass is
larger. The stated sensitivity of the balance is <0.1 μg with a weighing accuracy of ±0.1%, a
weighing precision of ±0.01%, and a signal resolution of <0.01 μg. The 24-h baseline drift is
stated to be <5 μg for an empty metalized quartz pan at 25 $^{\circ}$C and 20% RH. This is equivalent
to a relative mass change of <0.05% if the sample mass is 10 mg. As shown Section 3.3,
experimental tests do suggest that such performance can be reached.
**2.1.2 Humidity chamber**
The humidity chamber is used to regulate the temperature and RH under which
hygroscopicity and/or water adsorption of particles are investigated. Inside the humidity
chamber are housed a reference chamber (in which an empty pan is connected to the balance)
and a sample chamber (in which a sample pan is connected to the balance). A dry $N_2$ flow
(regulated by MFC2) is delivered through a humidifier and then mixed the second dry $N_2$ flow
(regulated by MFC1). The total flow is set to 200 sccm, and the ratio of these two flows can be
adjusted in order to regulate the final RH. After mixing, the flow is then split into two flows,
one delivered into the reference chamber and the other into the sample chamber. Therefore,
both chambers should have the same temperature and RH. Temperature inside the humidity
chamber can be adjusted from 5 to 85 $^{\circ}$C with a stated stability of ±0.1 $^{\circ}$C, and RH can be
varied between 0 and 98%. High accuracy in RH control, with a stated absolute accuracy of
±1%, is achieved by precisely controlling the dry and humidified $N_2$ flow rates, using mass
flow controllers regularly calibrated. The accuracy of RH control is routinely checked by
measurement of the DRH of NaBr, as detailed in Section 3.1 In addition, as shown in Figure 1,
two capacitance RH sensors are used to check relative humidity in the chamber.
The main advantage of using a reference chamber and a sample chamber is that the amount
of water adsorbed by the empty pan and the attached wire can be simultaneously determined
(and automatically subtracted using the provided software) under the same condition when
water uptake by particles under investigation is being measured. In addition, the effect of
buoyancy, which varies with RH (Beyer et al., 2014; Schroeder and Beyer, 2016), is also
automatically taken into account by using an empty pan as the reference. Semispherical quartz
crucibles with a volume of 180 μL, provided by the manufacturer, are used in this work as
sample pans.
**2.1.3 Other features**
Q5000SA is equipped with a programmable autosampler designed to deliver sample pans
into the humidity chamber. The autosampler can host up to 10 sample pans; however, in order
to minimize contamination by lab air, only one sample pan is uploaded into the autosampler
immediately prior to the measurement. The instrument status is displayed on a touch screen for
local operation. Q5000SA can also communicate with a computer via Ethernet. Two software
packages are provided by the manufacturer: 1) TA Instrument Explorer Q Series is used to
control the instrument, program measurement procedures, and log experimental data; 2) TA
Universal Analysis can be used for graphing experimental data in real time, data analysis, and
exporting data. Experimental data can be sampled with frequencies up to 1 Hz.

## 2.2 Experimental procedures

In our work two major experimental protocols are developed to 1) determine the DRH and 2) quantify water adsorption and/or mass hygroscopic growth. Corresponding experimental procedures are detailed below.

### 2.2.1 DRH determination

Based on the standard recommended by American Society for Testing and Materials International (ASTM, 2007) and TA Instruments (Waguespack and Hesse, 2007), an experimental method has been developed in this work to determine the DRH of a given sample, and it consists of the following steps. After the sample pan is properly located in the humidity chamber, temperature is set to the given value. After temperature is stabilized, RH is set to a value which is ~5% (when change/difference in RH is mentioned in this work, it always means the absolute value) higher than the anticipated DRH and the system is equilibrated for 120 min. For example, the DRH of NaBr is expected to be around (57-58)%, and RH was set to 62% from 0 to 120 min, as shown in Figure 2a. In the last step, RH is linearly decreased with a rate of 0.2% per min to a value which is ~5% lower than the anticipated DRH. For example, as shown in Figure 2a, RH was decreased from 62% at 120 min to 54% at 160 min, and the RH decrease rate was 0.2% per min.

Figure 2a shows changes of RH and sample mass (normalized to that at 0 min) as a function of time in an experiment to measure the DRH of NaBr at 25 ℃. RH was kept at 62% in the first 120 min during which the sample mass increased with time. After that, RH was linearly decreased to 54% with a rate of 0.2% RH per min, and during this period the sample mass first continued to increase to a maximum value and then decreased with time. The sample mass in the second period (120-160 min, as shown in Figure 2a) is plotted as a function of RH, and the RH (57.6% in this case) at which the sample mass reached the maximum value is equal to the measured DRH (ASTM, 2007). Measurement of a DRH usually takes ~3 h in total, and

experimental data such as RH and sample mass are recorded with a time resolution of 10 s. If
the DRH is unknown, we can increase the upper RH and decrease the lower RH used in the
measurement so that the DRH falls into the RH range. Increasing RH range used in the
measurement will of course lead to the increase of experimental time required.

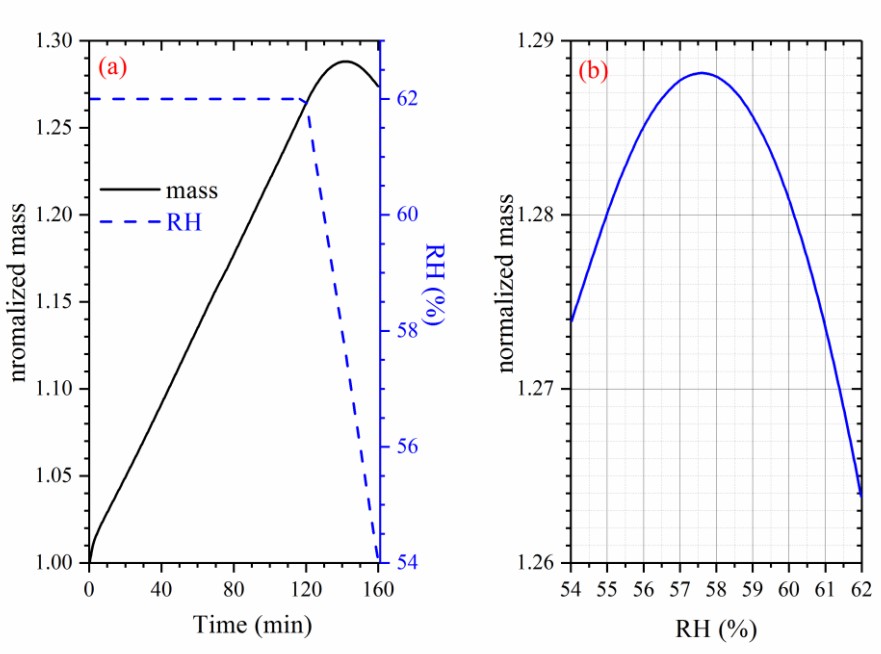


**Figure 2.** Typical experimental data in determination of DRH at a given temperature (NaBr at
25 °C as an example) using Q5000SA by linearly decreasing RH. (a): Change of normalized
sample mass (solid curve, left y-axis) and RH (dashed curve, right y-axis) as a function of time.
(b): Change of normalized sample mass as a function of RH when RH decreased linearly from
62% to 54%.

A second method has been developed to measure DRH at a given temperature. The particle
sample is first dried at 0% RH until its relative mass change is <0.05% within in 30 min. RH
is then increased to a value which is typically (5-10)% lower than the expected DRH and kept
at this level for 30 min. After that, RH is stepwise increased with an increment of 1% per step,
and at each step RH stays constant for 30 min. The DRH is equal to the RH at which a
significant increase in sample mass is first observed. Figure 3 shows changes in RH and
normalized sample mass as a function of time in an experiment designed to measure the DRH
of $(NH_4)_2SO_4$ at 25 °C. As shown in the shadowed region in Figure 3, a sharp increase in
normalized sample mass was first observed when RH was increased from 79% to 80%,
suggesting that deliquescence of $(NH_4)_2SO_4$ occurred between 79% and 80% RH at 25 °C.
Further increase in RH from 80% to 81% would cause sharper increase in sample mass,
confirming that deliquescence indeed occurred. DRH values measured by the two methods
agree with each other, and the second method is preferred because the occurrence of
deliquescence can be easily visualized from the experimental data.

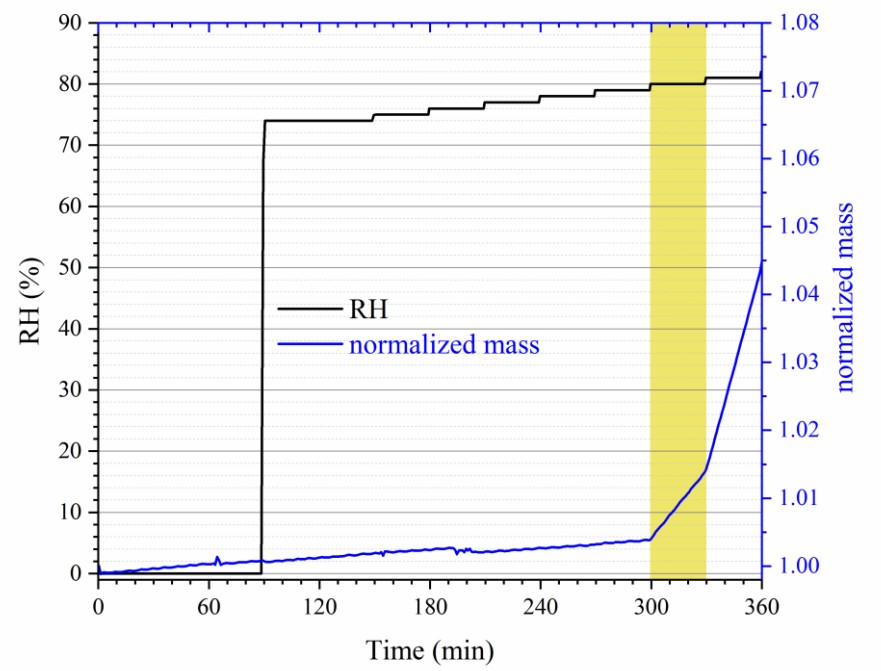


**Figure 3.** Typical experimental data in determination of DRH at a given temperature by
stepwise increasing RH. The experiment displayed in this figure was conducted to measure
DRH of $(NH_4)_2SO_4$ at 25 °C.
**2.2.2 Quantification of water adsorption and/or mass hygroscopic growth**
The following experimental procedures are used to determine the amount of water
adsorbed by a material (i.e. mass hygroscopic growth factors): 1) a sample pan is delivered into
the humidity chamber and temperature in the humidity chamber is set to a given value; 2) after
temperature becomes stable, RH in the humidity chamber is set to 0% and the sample is
equilibrated with the environment until its mass change is <0.05% within 30 min; 3) RH is
increased to a given value and the sample is equilibrated with the environment again until its
mass change is smaller than a certain value (typically 0.05% for less hygroscopic materials
such as $CaCO_3$ and fresh soot, and 0.1% for more hygroscopic materials such as $(NH_4)_2SO_4$
and NaCl) within 30 min; 4) RH is further increased to another given value and the sample is
equilibrated with the environment. The following assumptions are made to convert the mass of
adsorbed water to its surface coverage (Tang et al., 2016): 1) particles are spherical, having a
uniform diameter of 1 μm and a density of 2.5 g cm$^{-3}$, and 2) the average surface area that an
adsorbed water molecule occupies is $1\times10^{-15}$ cm$^2$. Under these assumptions, a mass change of
0.05% (relative to the dry mass) due to adsorption of water is equal to a surface coverage of
0.7 monolayers for adsorbed water.
All the processes are programmed, with the flexibility to choose the number of RH steps
and the corresponding RH values. Experimental data such as RH and sample mass are recorded
with a time resolution of 30 s. Relevant experimental results will be presented and discussed
in Sections 3.3 and 3.4.
**2.3 Chemicals**
Sodium bromide, provided by TA Instruments as a reference material for RH calibration,
was supplied by Alfa Aesar with a stated purity of >99.7%. Ammonium sulfate
(purity: >99.0%), sodium chloride (purity: >99.5%), potassium chloride (purity: >99.5%),
magnesium nitrate hexahydrate (purity: >99.0%), magnesium chloride hexahydrate
(purity: >99.0%), calcium bromide (purity: >99.98%), and calcium sulfate dihydrate
(purity: >99%) were purchased from Sigma-Aldrich. All the chemicals were used without
further pretreatment.

## 3 Results and Discussion

### 3.1 RH calibration

RH in vapor sorption analyzers and/or thermogravimetric analyzers can be calibrated/verified by determining the DRH of a reference material with a well-defined DRH (ASTM, 2007). In this work, NaBr provided by TA Instruments is used as the reference material (Waguespack and Hesse, 2007). We compare our measured DRHs of NaBr at six different temperatures with those reported by a previous study (Greenspan, 1977). The results are summarized in Table 1, suggesting that the differences between our measured and previous reported DRHs is <1% RH for temperatures ranging from 5 to 30 °C. The agreement is excellent, especially considering that 1) the RH has a stated accuracy of ±1% for our instrument and 2) DRH values reported by Greenspan (1977) typically have errors of ±0.5%. Further inspection of results compiled in Table 1 reveals that the difference is larger at lower temperature and becomes smaller at higher temperature.

**Table 1.** Comparison of DRHs (in %) of NaBr at different temperatures measured in our study with those reported in literature (Greenspan, 1977). The uncertainties for our measured DRH values are estimated to be ±1%.

| T (°C) | 5 | 10 | 15 | 20 | 25 | 30 |
|---|---|---|---|---|---|---|
| DRH (literature) | 63.5±0.7 | 62.2±0.6 | 60.7±0.5 | 59.1±0.4 | 57.6±0.4 | 56.0±0.4 |
| DRH (this work) | 62.2 | 61.2 | 60.0 | 58.5 | 57.6 | 56.1 |
| difference in DRH | 1.3 | 1.0 | 0.7 | 0.6 | 0.0 | -0.1 |

DRH values reported by Greenspan (1977), widely accepted as standard values, are recommended by the instrument manufacturer (Waguespack and Hesse, 2007) and also used in this study to calibrate our measure RH by taking into account the difference between our measured DRHs and those reported by Greenspan (1977) for NaBr at different temperatures.

All the RHs reported in this work (except measured DRHs of NaBr listed in Table 1) have been
calibrated. In our work we have not verified RH for temperature higher than 30 $^{\circ}$C because the
atmospheric relevance is limited. It should be pointed out if necessary, RH calibration can also
be carried out at higher temperature (up to 85 $^{\circ}$C) using the same procedure.
**3.2 DRH measurements**
Using the experimental method detailed in Section 2.2.1, we have measured DRHs of
$CaBr_2$, $MgCl_2 \cdot 6H_2O$, $Mg(NO_3)_2 \cdot 6H_2O$, NaCl, $(NH_4)_2SO_4$ and KCl at different temperatures
from 5 to 30 $^{\circ}$C. All the experimental results are summarized in Table 2. Figure 4a displays our
measured DRHs of these compounds at 25 $^{\circ}$C. DRHs range from ~20% to almost 90% for these
six compounds. As evident from Figure 4a, our measured DRHs show excellent agreement
with those reported by a previous study (Greenspan, 1977). Figure 4b shows the comparison
of our measured DRHs with those reported in literature for $Mg(NO_3)_2 \cdot 6H_2O$ as a function of
temperature (5-30 $^{\circ}$C), and excellent agreement is found again. It also appears that the
difference between our measured and previously reported DRH of $Mg(NO_3)_2 \cdot 6H_2O$ may show
a dependence on temperature; however, the difference is not significant compared to
uncertainties in DRH measurement. Careful examination of data compiled in Table 2 suggests
that the absolute difference between our measured and previously reported DRHs is typically
<1%.

**Table 2.** Comparison of DRHs measured by our study with those reported in literature
(Greenspan, 1977) for $CaBr_2$, $MgCl_2 \cdot 6H_2O$, $Mg(NO_3)_2 \cdot 6H_2O$, NaCl, $(NH_4)_2SO_4$ and KCl from
5-30 $^{\circ}$C. NA: data are not available. The uncertainties for our measured DRH values are
estimated to be ±1%.

| T ($^{\circ}$C) | literature | this work | literature | this work | literature | this work |
|---|---|---|---|---|---|---|
| | $CaBr_2$ | | $MgCl_2 \cdot 6H_2O$ | | $Mg(NO_3)_2 \cdot 6H_2O$ | |
| 5 | NA | 22.9 | 33.6±0.3 | 33.3 | 58.9±0.5 | 58.2 |

| | | | | | | |
|---|---|---|---|---|---|---|
| 10 | 21.6±0.5 | 21.6 | 33.5±0.3 | 33.9 | 57.4±0.4 | 57.1 |
| 15 | 20.2±0.5 | 20.0 | 33.3±0.3 | 33.6 | 55.9±0.3 | 55.6 |
| 20 | 18.5±0.5 | 18.0 | 33.1±0.2 | 33.5 | 54.4±0.3 | 54.4 |
| 25 | 18.5±0.5 | 17.1 | 32.8±0.2 | 33.2 | 52.9±0.3 | 53.1 |
| 30 | NA | 17.7 | 32.4±0.2 | 33.6 | 51.4±0.3 | 51.8 |
| | NaCl | | $(NH_4)_2SO_4$ | | KCl | |
| 5 | 75.6±0.5 | 76.0 | 82.4±0.7 | 80.8 | 87.7±0.5 | 86.7 |
| 10 | 75.7±0.4 | 75.7 | 82.1±0.5 | 80.8 | 86.8±0.4 | 86.3 |
| 15 | 75.6±0.3 | 75.7 | 81.7±0.4 | 80.4 | 85.9±0.4 | 85.6 |
| 20 | 75.5±0.3 | 75.6 | 81.3±0.3 | 80.3 | 85.1±0.3 | 85.0 |
| 25 | 75.3±0.3 | 75.2 | 81.0±0.3 | 79.6 | 84.3±0.3 | 83.9 |
| 30 | 75.1±0.3 | 75.5 | 80.6±0.3 | 79.7 | 83.6±0.3 | 83.4 |

In addition, we repeated the measurements of the DRH of $(NH_4)_2SO_4$ at 25 °C on several different days, and in total eight measurements have been carried out. The measured DRHs range from 79.5% to 80.1%. Therefore, it can be concluded from our systematical tests that the experimental method developed in our work can reliably measure DRHs from 5 to 30 °C.

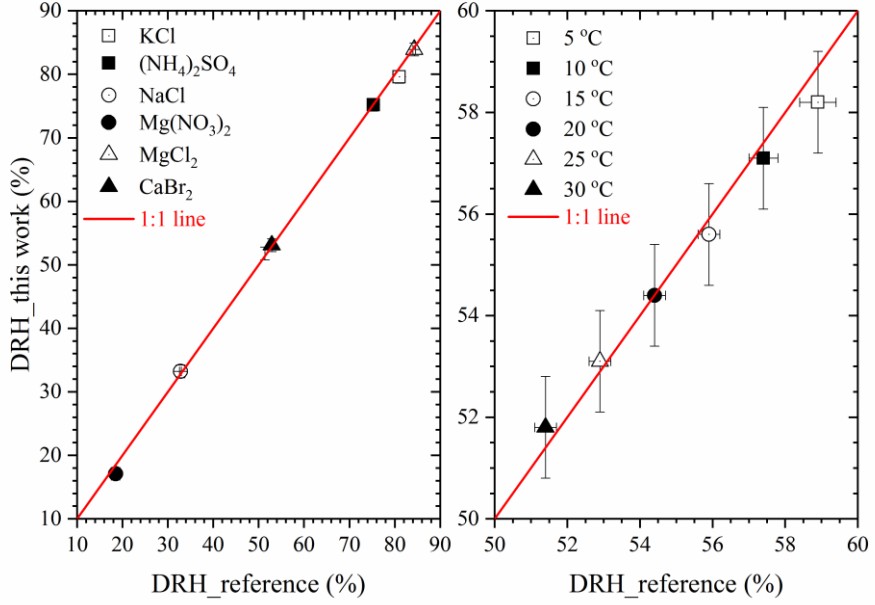

**Figure 4.** Comparison of our measured and previous reported DRHs (Greenspan, 1977). (a) DRHs of $(NH_4)_2SO_4$, NaCl, $MgCl_2·6H_2O$, $Mg(NO_3)_2·6H_2O$, $CaBr_2$ and KCl at 25 °C. Please

note that error bars are included, but they are too small to be clearly visible. (b) DRHs of
Mg(NO$_3$)$_2$·6H$_2$O as a function of temperature from 5-30 ℃.

**3.3 Mass hygroscopic growth measurements**

(NH$_4$)$_2$SO$_4$ and NaCl are important components found in tropospheric aerosol particles,
and their hygroscopicity has been well understood. They have also been widely used as
standard materials for validation of hygroscopicity and cloud condensation nucleation activity
measurements (Good et al., 2010; Ma et al., 2010b; Tang et al., 2015). In our work we have
measured mass hygroscopic growth factors of (NH$_4$)$_2$SO$_4$ and NaCl as a function of RH at two
different temperatures, with the purpose to further assess the performance of our instrument.
The mass hygroscopic growth factor is defined as the mass ratio of particles under dry
conditions to those at a given RH (Lee et al., 2008; Pope et al., 2010). Figure 5 show the
comparison of mass hygroscopic growth factors measured by our work with those predicted by
the E-AIM model (Wexler and Clegg, 2002). The agreement between measured and calculated
growth factors is excellent for NaCl at both temperatures; for (NH$_4$)$_2$SO$_4$, the agreement is not
as good as NaCl. This may be caused by two reasons. First, after (NH$_4$)$_2$SO$_4$ is deliquesced,
mass hygroscopic growth factors increase sharply with RH, and therefore a small difference in
RH would lead a relatively large change in measured mass hygroscopic growth factors; if
taking into account the uncertainty in RH (±1 %), the difference between our measured and
predicted mass hygroscopic growth factors is <15%. Second, inspection of the data in Table 2
reveals that the difference between our measured and previously reported DRH is <1% for all
the other compounds except (NH$_4$)$_2$SO$_4$. This may indicate that the purity of (NH$_4$)$_2$SO$_4$ could
lead to the small but yet detectable difference. In the near future we will purchase (NH$_4$)$_2$SO$_4$
with higher purity and measure its DRH and hygroscopic growth factors. Overall, it can be
concluded from the comparison that our measured mass hygroscopic growth factors agree well
with theoretical values for (NH$_4$)$_2$SO$_4$ and NaCl at both 5 and 25 ℃. This gives us further
confidence that the method developed in this work is reliable for hygroscopicity measurements
of atmospheric particles.

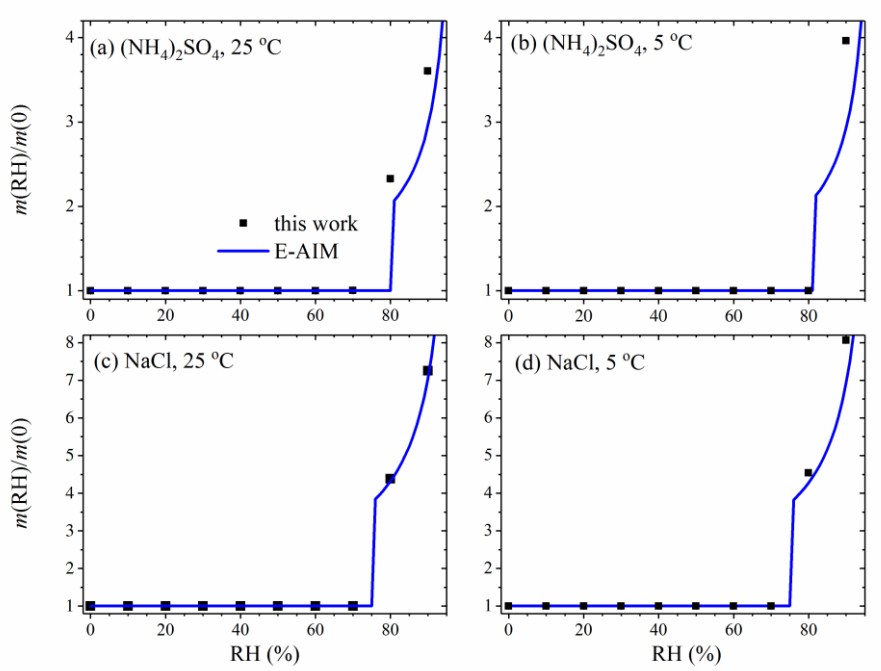


**Figure 5.** Comparison of mass hygroscopic growth factors measured in this work with these
predicted by the E-AIM model. (a) $(NH_4)_2SO_4$ at 5 °C; (b) $(NH_4)_2SO_4$ at 25 °C; (c) NaCl at
25 °C; (d) NaCl at 5 °C. Please note that error bars are included, but they are too small to be
clearly visible.

We have also measured the mass hygroscopic growth factors of $CaSO_4 \cdot 2H_2O$ as a function
of RH (up to 95%) at 25 °C. The results are plotted in Figure 6a, and the numerical data are
summarized in the appendix (Table A1). As shown in Figure 6, the ability of $CaSO_4 \cdot 2H_2O$ to
uptake water is very limited, with the mass ratio of adsorbed water to dry particles determined
to be (0.450±0.004)% (1 σ) at 95% RH. This is qualitatively consistent with two previous
studies which suggested that the cloud condensation nucleation activity of calcium sulfate
aerosol particles is very low (Sullivan et al., 2009a; Tang et al., 2015). More detailed
comparison and discussion are beyond the scope of this paper and will be addressed in a
following publication. This also implies that airborne $CaSO_4 \cdot 2H_2O$ particles are not
deliquesced for RH up to 95% and therefore may exist as non-spherical particles.

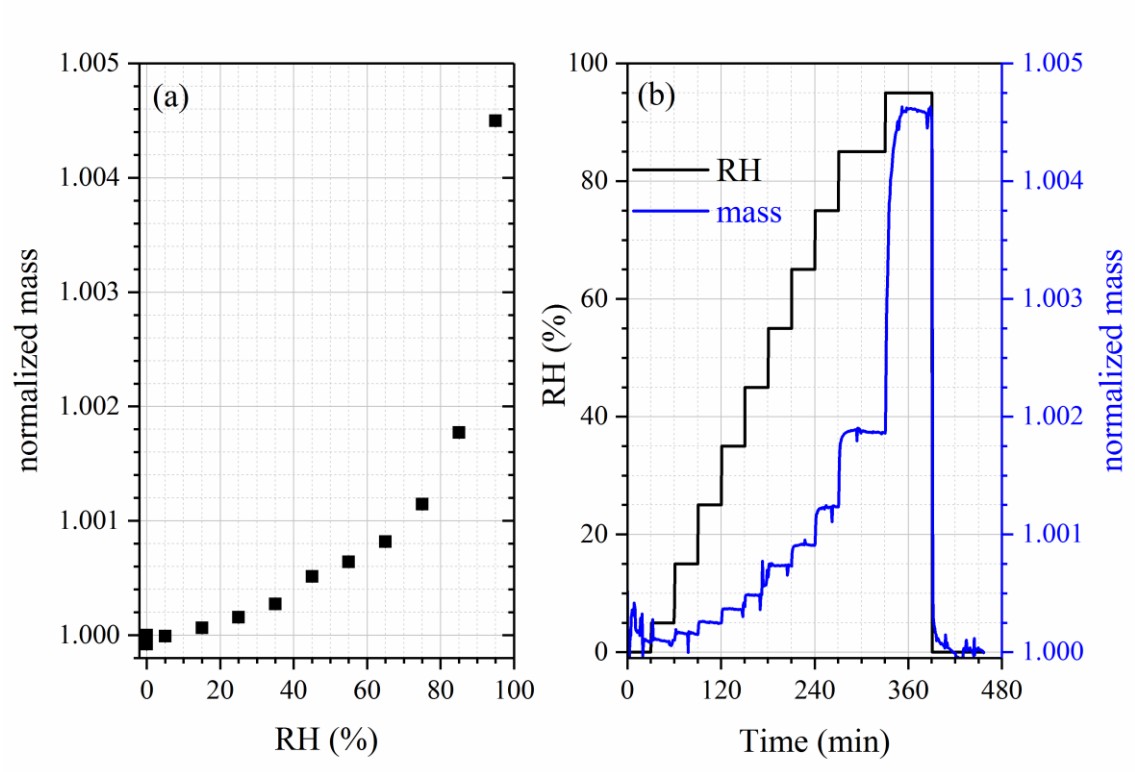

**Figure 6.** (a) Measured mass hygroscopic growth factors (normalized to the mass at 0% RH)
of $CaSO_4 \cdot 2H_2O$ as a function of RH up to 95% RH. Please note that error bars are included,
but they are too small to be clearly visible. (b) Time series of RH and normalized mass of
$CaSO_4 \cdot 2H_2O$ particles with a dry mass of ~9.05 mg during a hygroscopic growth experiment.
This experiment was carried out at 25 °C.

Figure 6b displays change of RH and normalized sample mass with time during the
measurement, suggesting that within 6 h our method can measure a relative mass change of
<0.025%. The accuracy of mass measurement is mainly limited by long-term baseline drifts.
In another experiment, a $CaCO_3$ sample (with a dry mass of ~10 mg) was used, and its mass
was continuously monitored under dry conditions at 25 °C. Under this experimental condition,
the baseline drift was determined to be <0.05% within 24 h.

## 4. Conclusion and outlook

The ability to uptake water vapor under subsaturated conditions is one of the most important physicochemical properties of atmospheric particles, largely determining their impacts on atmospheric chemistry and climate. In this work, we have developed a new experimental method to investigate interactions of particles with water vapor under subsaturated conditions at different temperatures from 5 to 30 $^{\circ}$C, using a commercial vapor sorption analyzer. Operation temperature can be increased up to 85 $^{\circ}$C, though the atmospheric relevance is limited. We have provided a detailed description of instrument configuration as well as experimental procedures to determine DRHs and mass hygroscopic growth factors. For the temperature range we have covered in this work (5-30 $^{\circ}$C), our measured DRHs of six different compounds with DRHs ranging from ~20% to ~90%, show excellent agreement with those reported in literature. In addition, mass hygroscopic growth factors measured in our work at different RH values agree well with those predicted by the E-AIM model for $(NH_4)_2SO_4$ and NaCl at 5 and 25 $^{\circ}$C. Therefore, we have demonstrated that experimental methods developed in our work can reliably measure DRHs and mass hygroscopic growth factors from 5 to 30 $^{\circ}$C.

To test the ability of this instrument to measure hygroscopic growth of compounds with low hygroscopicity, we have determined mass hygroscopic growth factors of $CaSO_4 \cdot 2H_2O$ at 25 $^{\circ}$C. It has been found that the ability of $CaSO_4 \cdot 2H_2O$ to uptake water is very limited. The mass of water adsorbed by $CaSO_4 \cdot 2H_2O$ at 95% RH is only (0.450±0.004)% of its dry mass. It has also been observed that this instrument can measure a mass change of <0.025% within 6 hours and <0.05% within 24 h, and accuracy of mass change determination is mainly limited by baseline drifts. With such an accuracy, this instrument is particularly useful for quantitative determination of water adsorption and/or hygroscopicity of non-spherical particles such as mineral dust and soot. Atmospheric aging processes are known to alter water adsorption, hygroscopicity and cloud condensation nucleation activity of mineral dust and soot particles

(Kelly and Wexler, 2005; Laskin et al., 2005; Zhang et al., 2008; Sullivan et al., 2009b; Han et al., 2013; Denjean et al., 2015; Tang et al., 2016). In the future, this instrument will be used to investigate water adsorption and hygroscopicity of mineral dust and soot particles before and after chemical processing. We note that this technique also has a few drawbacks: 1) this technique cannot be used to examine supersaturated droplets or determine efflorescence relative humidities (ERH), due to the contact of particles with the sample pan; 2) substantial amount of particles, typically around or larger than 1 mg, are required by this technique, limiting its application to atmospheric particles even after they are collected (e.g., using a filter or an impactor plate); 3) the experiment is very time-consuming, and a typical experiment can take several hours and even a few days, depending on experimental conditions.

## Data availability

Experimental data presented in this work are available upon request (Mingjin Tang: mingjintang@gig.ac.cn).

## Acknowledgement

Financial support provided by Chinese National Science Foundation (grant No.: 91644106 and 41675120), Chinese Academy of Sciences international collaborative project (grant No.: 132744KYSB20160036) and State Key Laboratory of Organic Geochemistry (grant No.: SKLOGA201603A) is acknowledged. Mingjin Tang would like to thank the CAS Pioneer Hundred Talents program for providing a starting grant and Yongjie Li would like to acknowledge funding support of the Start-up Research Grant from University of Macau (SRG2015-00052-FST). The authors declare no competing financial interest.

# Appendix

**Table A1.** Normalized mass of $CaSO_4 \cdot 2H_2O$ as a function of RH at 25 $^\circ$C.

| RH (%) | 0 | 5 | 15 | 25 | 35 | 45 |
|---|---|---|---|---|---|---|
| normalized mass | 1.00000 | 0.99999 | 1.00006 | 1.00016 | 1.00027 | 1.00051 |
| Error | 0.00001 | 0.00002 | 0.00001 | 0.00001 | 0.00004 | 0.00005 |
| RH (%) | 55 | 65 | 75 | 85 | 95 | 0 |
| normalized mass | 1.00064 | 1.00082 | 1.00114 | 1.00177 | 1.00450 | 0.99992 |
| error | 0.00002 | 0.00001 | 0.00001 | 0.00001 | 0.00004 | 0.00001 |

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
