# Peer review of "Investigation of water adsorption and hygroscopicity of atmospherically relevant particles using a commercial vapor sorption analyzer"

_Atmospheric Measurement Techniques, 2017_

## Referee Comment (RC4)

This paper reports measurements of DRH on bulk samples of some atmospherically-relevant salts. The technique as described is novel and such measurements are of importance in atmospheric chemistry. This paper is suitable for publication in AMT after the authors address the following major issues:

1. Experimental: What is the sample mass needed for this technique? Does this potentially limit its atmospheric applications? If the sample were to be extracted from a filter, what would the mass need to be?

2. Results and Discussion: More details on the E-AIM model should be provided: why was it chosen? What were the parameters used?

3. Figure 3: The agreement between reference and measured DRH values is excellent for most salts shown in Figure 3. However, $(NH_4)_2SO_4$ seems to be an outlier. Similarly, the data for $(NH_4)_2SO_4$ does not fully agree with the model in Figure 4. Can authors comment on this discrepancy? Was there perhaps some contamination in the $(NH_4)_2SO_4$ sample specifically?

4. Figure 4: I agree with the other reviewers that more points are needed on these plots to fully appreciate how measurements compare to the model.

5. Figure 4 and others: All figures that show experimental measurements need error bars. Similarly, uncertainties should be reported on measured DRH values in Table 1.

---

## Referee Comment (RC1) · Anonymous Referee #1 · 20 Jun 2017

Water adsorption and hygroscopicity are among the most important physicochemical properties of aerosol particles. The authors developed a novel method to provide the information on mass hygroscopic growth of atmospheric particles. It can be considered to be published in AMT after modifications. (1) There is no information on how to collect the samples on the pan. The effects of the weight and thickness of dry sample on the results should be discussed. (2) Page 8, 169: determine the DRH. In this section, the authors mentioned that the method was developed based on the ASTM, 2007, but, lack of the detail description on the principles. The authors should descript why the DRH can be determined by following step 1)-3) (Line 176-179). Typically, the efflorescence RH was detected by measuring the change in hygroscopic growth with decreasing

[Figure]

RH. Why, DRH was detected by this method? (3) Page 8, 178-179: RH is set to a value which is ~5% (when change/difference in RH is mentioned in this work, it always means the absolute value) higher than the anticipated DRH. Here, "higher" or "lower"? (4) Page 9 180-181: 3) RH is linearly decreased with a rate of 0.2% per min to a value which is ~5% lower than the anticipated DRH. What does mean here? (5) Page 8 178-Page 9 181: The description is different from what were done in Figure 2. (6) In Figure 4: Too few data points are given. Only one data point showed the particle growth factor. It is difficult to judge the agreement is good or not. (7) If the slow response of vapor sorption analyzer (hours for each measurement) is a drawback for the future applications?
* * *

---

## Short Comment (SC1) · 30 Jul 2017

P15, Line 292-296ïïjŇFigure 4 The authors mentioned that the measured mass hygroscopic growth factors agree well with the E-AIM model for (NH4)2SO4 and NaCl at temperature of 25 and 5 degree centigrade. According to Figure 4, the data of NaCl measured agree well with model. But in Figure 4 (a) and (b), it seems that the (NH4)2SO4 data from this work are obviously higher than that from the model, conflicting with the author's description.

---

## Referee Comment (RC2) · Anonymous Referee #2 · 14 Aug 2017

**Summary:**

This work proposed a method to quantify water adsorption and mass hygroscopic growth of atmospheric particles. This new experimental method can be easily applied with a commercial vapor sorption analyzer. The manuscript fits well to the scope of AMT and presents valuable methods. Thus I recommend it to be published after the following comments listed below have been adequately addressed.

**Comments:**

1. Regarding to the title or the key points of this manuscript: Investigation of water adsorption and hygroscopicity of ***atmospheric particles*** using a commercial vapor sorption analyzer: the materials the author used in this study are not atmospheric particles, but actually bulk samples. Please clarify how these results represent atmospheric conditions.

2. In section 2.2.1 and 3.2, the authors demonstrate how to determine the DRH and the results seem agree well with the literature. I am wondering could you use the similar procedures to determine ERH?

3. In section 3.3, only ammonium sulfate and sodium chloride are atmospheric-relevant species. I would suggest the author also present the results for atmospheric-relevant compounds, for instance, organic compounds to represent low-hygroscopic species.

4. Please also give detailed definition of mass hygroscopic growth factor in your case as people are using hygroscopic growth factor directly to express hygroscopcity of particles when using e.g. HTDMAs.

5. How did the author come up with using E-AIM model? This is not a good choice to use to validate your results as E-AIM represents better for mixtures rather than single compound. Hence, I suggest the author either gives better description of E-AIM model to prove you have good knowledgement of this tool or compares the results from other techniques.

6. I would strongly suggest the author plot a complete humidograms of each compound measured by the vapor sorption analyzer, i.e. the mass growth factor as a function of different RH. The author stated that this commercial instrument provides a robust method to

investigate water adsorption and hygroscopicity. However, only DRH and one MGF point above DRH were reported. I do not think this is enough. Meanwhile, the reported MGF seems higher than the E-AIM results for AS. Please consider fulfilling the datasets and plotting similar figures as Fig. 3 and Fig. 4 for MGF values. Without the comparison of growth factor values from your instruments and other researches, it is difficult to validate your results.

7. There are several grammar mistakes in the text, please carefully check.

**References**

---

## Author Comment (AC5) · 7 Sep 2017

The comment was uploaded in the form of a supplement:
https://www.atmos-meas-tech-discuss.net/amt-2017-56/amt-2017-56-AC5-supplement.pdf

---

## Author Comment (AC6) · 7 Sep 2017

We would like to thank Professor Zhang for his comment on our manuscript. This issue has also been raised by other referees. We have addressed this comment in the revised manuscript. Please refer to our revised manuscript (line 326-340) and our reply to ref #3 and ref #4 for more details.

---

## Author Response (AR1)

Comments by Referees are in blue. Our replies are in black. Changes to the manuscript are highlighted in red both in here and in the revised manuscript.

**Reply to Ref #1**

Water adsorption and hygroscopicity are among the most important physicochemical properties of aerosol particles. The authors developed a novel method to provide the information on mass hygroscopic growth of atmospheric particles. It can be considered to be published in AMT after modifications.

**Author reply:** We would like to thank Ref #1 for his/her highly positive comments on our manuscript. All the comments have been properly addressed in our revised manuscript, as detailed below.

(1) There is no information on how to collect the samples on the pan. The effects of the weight and thickness of dry sample on the results should be discussed.

**Author reply:** In the revised manuscript **(line 136-140)** we have included a few sentences to explain how particle samples are prepared and to discuss the effects of sample mass: "Powdered particles are transferred into the sample pan using a small stainless-steel spatula. The mass of the sample would not affect the measured mass ratio of dry particles to associated water under a given condition; however, it would take more time to reach the equilibrium if the sample mass is larger."

(2) Page 8, 169: determine the DRH. In this section, the authors mentioned that the method was developed based on the ASTM, 2007, but, lack of the detail description on the principles. The authors should descript why the DRH can be determined by following step 1)-3) (Line 176-179). Typically, the efflorescence RH was detected by measuring the change in hygroscopic growth with decreasing RH. Why, DRH was detected by this method?

**Author reply:** We agree with the referee that the principle of this experimental method is not very clear. After we submitted this manuscript, we have adopted a second method to measure DRH. The second method measures the mass change when RH is stepwise increased with an increment of 1% per step. The second method is preferred (and will also be always used in future) because the occurrence of deliquescence can be detected at the first RH when a significant increase in sample mass is observed. In the revised manuscript **(line 215-232)**, we have added one paragraph and a new figure to describe this experimental method. Please refer to our revised manuscript for further details.

(3) Page 8, 178-179: RH is set to a value which is ~5% (when change/difference in RH is mentioned in this work, it always means the absolute value) higher than the anticipated DRH. Here, "higher" or "lower"?

(4) Page 9 180-181: 3) RH is linearly decreased with a rate of 0.2% per min to a value which is ~5% lower than the anticipated DRH. What does mean here?

(5) Page 8 178-Page 9 181: The description is different from what were done in Figure 2.

**Author reply:** The three comments above are all related to one experimental method, and they are addressed together. In the revised manuscript **(line 191-195)** we have rephrased several sentences in this paragraph to make our experimental procedure more clear: "After the sample pan is properly located in the humidity chamber, temperature is set to the given value. After temperature is stabilized, RH is set to a value which is ~5% (when change/difference in RH is mentioned in this work, it always means the absolute value) higher than the anticipated DRH and the system is equilibrated for 120 min. For example, the DRH of NaBr is expected to be around (57-58)%, and RH was set to 62% from 0 to 120 min, as shown in Figure 2a. In the last step, RH is linearly decreased with a rate of 0.2% per min to a value which is ~5% lower than the anticipated DRH.

For example, as shown in Figure 2a, RH was decreased from 62% at 120 min to 54% at 160 min, and the RH decrease rate was 0.2% per min."

(6) In Figure 4: Too few data points are given. Only one data point showed the particle growth factor. It is difficult to judge the agreement is good or not.

**Author reply:** We agree with the referee. We have conducted additional measurements with RH increment was reduced from 30% to 10%, and the new results have been presented. In our revised manuscript **(line 326-345)** we have also expanded our discussion on comparison of our measurement with E-AIM predictions.

(7) If the slow response of vapor sorption analyzer (hours for each measurement) is a drawback for the future applications?

**Author reply:** We agree with the referee. In the revised manuscript **(line 398-404)** we have discussed the drawbacks of this technique: "We note that this technique also has a few drawbacks: 1) this technique cannot be used to examine supersaturated droplets or determine efflorescence relative humidities (ERH), due to the contact of particles with the sample pan; 2) substantial amount of particles, typically around or larger than 1 mg, are required by this technique, limiting its application to atmospheric particles even after they are collected (e.g., using a filter or an impactor plate); 3) the experiment is very time-consuming, and a typical experiment can take several hours and even a few days, depending on experimental conditions."

Comments by Referees are in blue. Our replies are in black. Changes to the manuscript are highlighted in red both in here and in the revised manuscript.

**Reply to Ref #2**

Summary: This work proposed a method to quantify water adsorption and mass hygroscopic growth of atmospheric particles. This new experimental method can be easily applied with a commercial vapor sorption analyzer. The manuscript fits well to the scope of AMT and presents valuable methods. Thus I recommend it to be published after the following comments listed below have been adequately addressed.

**Author reply:** We would like to thank Ref #2 for his/her very valuable comments, which have significantly help improve our manuscript. All the comments have been properly addressed in our revised manuscript, as detailed below.

Comments: 1. Regarding to the title or the key points of this manuscript: Investigation of water adsorption and hygroscopicity of atmospheric particles using a commercial vapor sorption analyzer: the materials the author used in this study are not atmospheric particles, but actually bulk samples. Please clarify how these results represent atmospheric conditions.

**Author reply:** We agree with the referee that our original title does not precisely reflect what we did. Although particles used in our work are not airborne, these materials, such as $(NH_4)_2SO_4$ and NaCl, are commonly found in atmospheric particles and thus relevant for the atmosphere.  In the revised manuscript we have changed the title to "Investigation of water adsorption and hygroscopicity of atmospherically relevant particles using a commercial vapor sorption analyzer".

2. In section 2.2.1 and 3.2, the authors demonstrate how to determine the DRH and the results seem agree well with the literature. I am wondering could you use the similar procedures to determine ERH?

**Author reply:** Unfortunately ERH cannot be determined using this technique. In the revised manuscript (line 398-404) we have discussed the drawbacks of this technique: "We note that this technique also has a few drawbacks: 1) this technique cannot be used to examine supersaturated droplets or determine efflorescence relative humidities (ERH), due to the contact of particles with the sample pan; 2) substantial amount of particles, typically around or larger than 1 mg, are required by this technique, limiting its application to atmospheric particles even after they are collected (e.g., using a filter or an impactor plate); 3) the experiment is very time-consuming, and a typical experiment can take several hours and even a few days, depending on experimental conditions."

3. In section 3.3, only ammonium sulfate and sodium chloride are atmospheric-relevant species. I would suggest the author also present the results for atmospheric-relevant compounds, for instance, organic compounds to represent low-hygroscopic species.

**Author reply:** In addition to $(NH_4)_2SO_4$ and NaCl, we have also measured DRH of several Ca-containing and Mg-containing compounds and the mass hygroscopic growth factors of $CaCO_4 \cdot 2H_2O$. These compounds can be formed in the troposphere due to heterogeneous reactions of calcite and dolomite, contained in mineral dust, with acidic trace gases. In addition, as shown in our work, the hygroscopicity of $CaCO_4 \cdot 2H_2O$ is very low. Therefore, we believe that data presented in our manuscript is enough for a technical paper. As pointed out by the referee, this technique should be very useful to study organic particles with low hygroscopicity, and this is indeed a field our technique will be applied to.

4. Please also give detailed definition of mass hygroscopic growth factor in your case as people are using hygroscopic growth factor directly to express hygroscopicity of particles when using e.g. HTDMAs.

**Author reply:** In the revised manuscript **(line 323-324)**, we have added one sentence to define the mass hygroscopic growth factor: "The mass hygroscopic growth factor is defined as the mass ratio of particles under dry conditions to those at a given RH (Lee et al., 2008; Pope et al., 2010)."

5. How did the author come up with using E-AIM model? This is not a good choice to use to validate your results as E-AIM represents better for mixtures rather than single compound. Hence, I suggest the author either gives better description of E-AIM model to prove you have good knowledgement of this tool or compares the results from other techniques.

**Author reply:** In fact the E-AIM model has been widely accepted and used, and many previous studies compared their measured hygroscopic growth factors of NaCl and (NH4)$_2$SO$_4$ to these predicted by the E-AIM model, in order to verify their measurements. In the revised manuscript **(line 103-108)** we have included a few sentences to explain why this model is used in our work, and to refer readers to original papers for more details of this model: "Detailed description of the E-AIM model can be found elsewhere (Clegg et al., 1998; Friese and Ebel, 2010). Hygroscopic growth factors, calculated using the E-AIM model, has been widely used to compare with experimental measurements to verify the performance of a variety of instruments, techniques and/or methods developed for hygroscopic growth studies (Pope et al., 2010; Lei et al., 2014; Estillore et al., 2016)."

6. I would strongly suggest the author plot a complete humidograms of each compound measured by the vapor sorption analyzer, i.e. the mass growth factor as a function of different RH. The author stated that this commercial instrument provides a robust method to investigate water adsorption and hygroscopicity. However, only DRH and one MGF point above DRH were reported. I do not think this is enough. Meanwhile, the reported MGF seems higher than the E-AIM results for AS. Please consider fulfilling the datasets and plotting similar figures as Fig. 3 and Fig. 4 for MGF

values. Without the comparison of growth factor values from your instruments and other researches, it is difficult to validate your results.

**Author reply:** We agree with the referee. We have conducted additional measurements with RH increment was reduced from 30% to 10%, and the new results have been presented. We have also expanded our discussion on comparison of our measurements with E-AIM predictions. Please refer to our revised manuscript **(line 326-345)** for more details.

7. There are several grammar mistakes in the text, please carefully check.

**Author reply:** As suggested, we have checked our revised manuscript carefully and thoroughly.

Comments by Referees are in blue. Our replies are in black. Changes to the manuscript are highlighted in red both in here and in the revised manuscript.

**Reply to Ref #3**

This manuscript describes a technique for determining the mass growth factor (MGF) and deliquescence relative humidity (DRH) of compounds using a commercial vapour sorption analyser. The authors present the methodology for determining the DRH and the MGF, followed by measurements of DRHs for different compounds/temperatures intended to confirm the RH calibration of the instrument. They then present measurements of the mass hygroscopic growth of ammonium sulphate and sodium chloride at two different temperatures, followed by calcium sulphate dihydrate which is used as a mimic for a low hygroscopicity species. In Lines 94-97, the authors note that two groups have already reported measurements using a similar technique, but that the novelty of their manuscript lies in the fact that it systematically evaluates the performance of the technique which has not been done before. I believe the manuscript is suitable for publication in AMT only after the concerns regarding the rigour of this validation included in the list below are adequately addressed.

**Author reply:** We would like to thank Ref #3 for his/her very valuable comments, which have significantly help improve our manuscript. All the comments have been properly addressed in our revised manuscript, as detailed below.

(Line 1-2) The title of the manuscript is factually incorrect. The paper reports water adsorption and hygroscopicity of atmospherically relevant compounds, not of atmospheric particles (i.e. sampled directly from the atmosphere). This should be changed.

**Author reply:** We agree with the referee that our original title does not precisely reflect what we did. Although particles used in our work are not airborne, these materials, such as $(NH_4)_2SO_4$ and

NaCl, are commonly found in atmospheric particles and thus relevant for the atmosphere. In the revised manuscript we have changed the title to "Investigation of water adsorption and hygroscopicity of atmospherically relevant particles using a commercial vapor sorption analyzer". (Lines 67-72) The authors say the strength of the technique is the ability to make measurements on non-spherical particles, which can be problematic for some of the more commonly used techniques which make measurements on species in the aerosol phase. However, the authors do not mention any of the drawbacks of looking at the hygroscopic behaviour of particles on a hydrophilic surface, in that they will not be able to access supersaturated solute states or determine the efflorescence RH.

**Author reply:** The referee is right. In the revised manuscript **(line 398-404)** we have discussed the drawbacks of this technique: "We note that this technique also has a few drawbacks: 1) this technique cannot be used to examine supersaturated droplets or determine efflorescence relative humidities (ERH), due to the contact of particles with the sample pan; 2) substantial amount of particles, typically around or larger than 1 mg, are required by this technique, limiting its application to atmospheric particles even after they are collected (e.g., using a filter or an impactor plate); 3) the experiment is very time-consuming, and a typical experiment can take several hours and even a few days, depending on experimental conditions."

(Line 155-156) The authors state that the RH can be varied between 0 and 98% with an absolute accuracy of +/- 1% as measured by a probe. Can the authors provide any details on the type of probe used here? Capacitance probes typically used for RH determination can be expected to have an accuracy of +/- 2% at RHs below 80%, but the error can climb to +/- 3% at RHs above this. These are obviously significantly larger than the quoted value.

**Author reply:** The referee has made a good point here. In our original manuscript we did make it very clear how to control and determine RH. In fact, high accuracy of RH control is achieved by precise control of the dry and wet flow rates. In the revised manuscript **(line 156-160)** we have added a few sentences to clarify it: "High accuracy in RH control, with a stated absolute accuracy of $\pm 1\%$, is achieved by precisely controlling the dry and humidified $N_2$ flow rates, using mass flow controllers regularly calibrated. The accuracy of RH control is routinely checked by measurement of the DRH of NaBr, as detailed in Section 3.1 In addition, as shown in Figure 1, two capacitance RH sensors are used to check relative humidity in the chamber."

(Line 177-181) If the authors are measuring the DRH why does the experimental method involve setting the RH higher than the DRH and then lowering it slowly? If there is no lag between slowly changing the desired RH in the software and this value equilibrating in the chamber (the authors do not mention one), why can the measurement not be performed by increasing the RH slowly from a value below the DRH? This way you would get a lot more data which would actually visualise the step change in mass as the particle deliquesced.

**Author reply:** Indeed the method suggested by the referee has some advantage. We have also realized this after we submitted our manuscript, and DRH values have also been measured using this method, showing good agreement with those determined used the method described in our original manuscript. In the revised manuscript **(line 215-232)**, we have added one paragraph and a new figure to describe this experimental method. Please refer to our revised manuscript for further details.

Figure 2. Error bars in the RH (from the absolute accuracy of the probe stated in line 156) need adding to this plot.

**Author reply:** This figure shows the raw data without being processed, and therefore it is not necessary to have the error bars. However, as suggested by the referees, error bars have been added for other figures.

Table 1 and Table 2. DRH values determined from this work need an associated error originating from the probe accuracy.

**Author reply:** In the revised manuscript uncertainty information has been provided for these two tables.

Figure 3. Errors bars in RH are included for the DRH reference data but not for the DRH values determined in this work. These need adding to the plot.

**Author reply:** In the revised manuscript we have provided uncertainty information for our measured DRH values displayed in this Figure. Please note that in some cases the error bars are too smaller to be clearly visible.

Also, the authors should comment on the systematic disagreement between the data as a function of temperature which can be seen in panel b).

**Author reply:** It looks like that there is systematic disagreement as a function of temperature, as pointed out by the referee; however, the difference is not significant compared to uncertainties in DRH measurement. In the revised manuscript **(line 295-298)** we have added one sentence to discuss this: "It also appears that the difference between our measured and previously reported DRH of $Mg(NO_3)_2 \cdot 6H_2O$ may show a dependence on temperature; however, the difference is not significant compared to uncertainties in DRH measurement."

Figure 4. There are not enough data points on the graphs here to use them as a validation of the technique. In each panel all but one of the data points are on the flat region of the hygroscopic curve and there are no data at all either side of the deliquescence event, which is actually the region

of interest. Further data points need adding to each plot which show a clear tracing out of the deliquescence region and more data points should also be added at high RH as this is the region most relevant to activation of aerosol in the atmosphere. Error bars in the RH (from the probe) should be included.

**Author reply:** We agree with the referee. We have conducted additional measurements with RH increment was reduced from 30% to 10%, and the new results have been presented with error bars included. We have also expanded our discussion on comparison of our measurement with E-AIM predictions. Please refer to our revised manuscript **(line 326-345)** for more details.

The authors should also comment on the fact that the ammonium sulphate data at both temperatures deviates from E-AIM at high RH.

**Author reply:** As suggested, in the revised manuscript **(line 326-340)** we have added a few sentences to discuss the issue raised by the referee: "The agreement between measured and calculated growth factors is excellent for NaCl at both temperatures; for $(NH_4)_2SO_4$, the agreement is not as good as NaCl. This may be caused by two reasons. First, after $(NH_4)_2SO_4$ is deliquesced, mass hygroscopic growth factors increase sharply with RH, and therefore a small difference in RH would lead a relatively large change in measured mass hygroscopic growth factors; if taking into account the uncertainty in RH ($\pm 1$ %), the difference between our measured and predicted mass hygroscopic growth factors is <15%. Second, inspection of the data in Table 2 reveals that the difference between our measured and previously reported DRH is <1% for all the other compounds except $(NH_4)_2SO_4$. This may indicate that the purity of $(NH_4)_2SO_4$ could lead to the small but yet detectable difference. In the near future we will purchase $(NH_4)_2SO_4$ with higher purity and measure its DRH and hygroscopic growth factors. Overall, it can be concluded from the comparison that our measured mass hygroscopic growth factors agree well with theoretical values

for $(NH_4)_2SO_4$ and NaCl at both 5 and 25 °C. This gives us further confidence that the method developed in this work is reliable for hygroscopicity measurements of atmospheric particles."

Figure 5 panel b) The raw mass data shows frequent dips and spikes. What is the origin of this?

**Author reply:** Because the mass change is very small, these dips in spikes are due to random noise in mass measurement. As we stated in the original manuscript, the noise level, smaller than 0.025%, determines our detection limit in hygroscopicity measurement.

(Line 314) Is this number supposed to be 0.025%?

**Author reply:** The referee is right. It should be 0.025%, and we have corrected it in the revised manuscript.

(General comment) The authors say the greatest advantage of this technique is the ability to look at non-spherical particles, however they report no measurements (even preliminary ones) of non-spherical particles here. The impact of this manuscript would be much higher if some were to be included.

**Author reply:** The principle of our technique relies on measurements of mass change and thus particles under investigation do not need to be spherical. Our work show that for RH in the range of 95%, $CaSO_4 \cdot 2H_2O$ particles are not deliquesced and thus may exist as non-spherical particles. In the revised manuscript **(line 355-356)**, we have added one sentence to make this more clear:

"This also implies that airborne $CaSO_4 \cdot 2H_2O$ particles are not deliquesced for RH up to 95% and therefore may exist as non-spherical particles."

Comments by Referees are in blue. Our replies are in black. Changes to the manuscript are highlighted in red both in here and in the revised manuscript.

**Reply to Ref #4**

This paper reports measurements of DRH on bulk samples of some atmospherically-relevant salts. The technique as described is novel and such measurements are of importance in atmospheric chemistry. This paper is suitable for publication in AMT after the authors address the following major issues:

**Author reply:** We would like to thank Ref #4 for his/her very valuable comments, which have significantly help improve our manuscript. All the comments have been properly addressed in our revised manuscript, as detailed below.

1. Experimental: What is the sample mass needed for this technique? Does this potentially limit its atmospheric applications? If the sample were to be extracted from a filter, what would the mass need to be?

**Author reply:** In our original manuscript (line 129-131) we have mentioned the mass range of particles used in our work. Indeed this limits the atmospheric application of the technique. In the revised manuscript **(line 398-404)** we have discussed drawbacks of this technique: "We note that this technique also has a few drawbacks: 1) this technique cannot be used to examine supersaturated droplets or determine efflorescence relative humidities (ERH), due to the contact of particles with the sample pan; 2) substantial amount of particles, typically around or larger than 1 mg, are required by this technique, limiting its application to atmospheric particles even after they are collected (e.g., using a filter or an impactor plate); 3) the experiment is very time-consuming, and a typical experiment can take several hours and even a few days, depending on experimental conditions."

Furthermore, in the revised manuscript **(line 136-140)** we have included a few sentences to explain how particle samples are prepared and to discuss the effects of sample mass: "Powdered particles are transferred into the sample pan using a small stainless-steel spatula. The mass of the sample would not affect the measured mass ratio of dry particles to associated water under a given condition; however, it would take more time to reach the equilibrium if the sample mass is larger."

2. Results and Discussion: More details on the E-AIM model should be provided: why was it chosen? What were the parameters used?

**Author reply:** We chose to use the E-AIM model because it is very user-friendly and has been widely accepted and used. We do not provide technical details of this model in our manuscript, but instead refer readers to original papers. In the revised manuscript **(line 103-108)** we have included a few sentences to explain why the E-AIM model is used in our work: "Detailed description of the E-AIM model can be found elsewhere (Clegg et al., 1998; Friese and Ebel, 2010). Hygroscopic growth factors, calculated using the E-AIM model, has been widely used to compare with experimental measurements to verify the performance of a variety of instruments, techniques and/or methods developed for hygroscopic growth studies (Pope et al., 2010; Lei et al., 2014; Estillore et al., 2016)."

3. Figure 3: The agreement between reference and measured DRH values is excellent for most salts shown in Figure 3. However, (NH4)2SO4 seems to be an outlier. Similarly, the data for (NH4)2SO4 does not fully agree with the model in Figure 4. Can authors comment on this discrepancy? Was there perhaps some contamination in the (NH4)2SO4 sample specifically?

**Author reply:** The referee raised a very good point. As suggested, in the revised manuscript **(line 326-340)** we have added a few sentences to discuss this issue: "The agreement between measured and calculated growth factors is excellent for NaCl at both temperatures; for $(NH_4)_2SO_4$, the

agreement is not as good as NaCl. This may be caused by two reasons. First, after $(NH_4)_2SO_4$ is deliquesced, mass hygroscopic growth factors increase sharply with RH, and therefore a small difference in RH would lead a relatively large change in measured mass hygroscopic growth factors; if taking into account the uncertainty in RH ($\pm1$ %), the difference between our measured and predicted mass hygroscopic growth factors is <15%. Second, inspection of the data in Table 2 reveals that the difference between our measured and previously reported DRH is <1% for all the other compounds except $(NH_4)_2SO_4$. This may indicate that the purity of $(NH_4)_2SO_4$ could lead to the small but yet detectable difference. In the near future we will purchase $(NH_4)_2SO_4$ with higher purity and measure its DRH and hygroscopic growth factors. Overall, it can be concluded from the comparison that our measured mass hygroscopic growth factors agree well with theoretical values for (NH4)2SO4 and NaCl at both 5 and 25 oC. This gives us further confidence that the method developed in this work is reliable for hygroscopicity measurements of atmospheric particles."

4. Figure 4: I agree with the other reviewers that more points are needed on these plots to fully appreciate how measurements compare to the model.

**Author reply:** As requested, we have conducted these experiments with a RH resolution of 10%, and discussion on the comparison between model prediction and our measurement has also been expanded. Please refer to our revised manuscript **(line 326-345)** for more details.

5. Figure 4 and others: All figures that show experimental measurements need error bars. Similarly, uncertainties should be reported on measured DRH values in Table 1.

**Author reply:** As requested, information on uncertainties has been provided for all the figures and tables. Please note that in some figures the error bars are too smaller to be clearly visible.